# Energy Efficiency for Green Internet of Things (IoT) Networks: A Survey †

Laith Farhan [1,2,*], Rasha Subhi Hameed [3], Asraa Safaa Ahmed [4], Ali Hussein Fadel [4], Waled Gheth [1], Laith Alzubaidi [5], Mohammed A. Fadhel [6] and Muthana Al-Amidie [7]

1   Faculty of Science and Engineering, Manchester Metropolitan University, Manchester M15 6BH, UK; waled-ali.m.gheth@stu.mmu.ac.uk
2   College of Engineering, University of Diyala, Baqubah 32001, Iraq
3   Department of Computer Science, College of Education for Pure Science, University of Diyala, Baqubah 32001, Iraq; purecomp.rasha.hameed@uodiyala.edu.iq
4   Department of Computer Science, College of Science, University of Diyala, Baqubah 32001, Iraq; asraasafaa@uodiyala.edu.iq (A.S.A.); ali_hussein_fadel@uodiyala.edu.iq (A.H.F.)
5   Faculty of Science & Engineering, Queensland University of Technology, Brisbane, QLD 4000, Australia; laith.alzubaidi@hdr.qut.edu.au
6   College of Computer Science & Information Technology, University of Sumer, Thi Qar 64005, Iraq; Mohammed.a.fadhel@uoitc.edu.iq
7   Faculty of Electrical Engineering & Computer Science, University of Missouri, Columbia, MO 65211, USA; mkidn6@mail.missouri.edu
*   Correspondence: l.al-bayati@mmu.ac.uk
†   This paper an extended version of our thesis published in Laith Kadhim Farhan (2020) Energy Efficiency in Green Internet of Things (IoT) Networks at Manchester Metropolitan University.

**Abstract:** The last decade has witnessed the rise of the proliferation of Internet-enabled devices. The Internet of Things (IoT) is becoming ever more pervasive in everyday life, connecting an ever-greater array of diverse physical objects. The key vision of the IoT is to bring a massive number of smart devices together in integrated and interconnected heterogeneous networks, making the Internet even more useful. Therefore, this paper introduces a brief introduction to the history and evolution of the Internet. Then, it presents the IoT, which is followed by a list of application domains and enabling technologies. The wireless sensor network (WSN) is revealed as one of the important elements in IoT applications, and the paper describes the relationship between WSNs and the IoT. This research is concerned with developing energy-efficiency techniques for WSNs that enable the IoT. After having identified sources of energy wastage, this paper reviews the literature that discusses the most relevant methods to minimizing the energy exhaustion of IoT and WSNs. We also identify the gaps in the existing literature in terms of energy preservation measures that could be researched and it can be considered in future works. The survey gives a near-complete and up-to-date view of the IoT in the energy field. It provides a summary and recommendations of a large range of energy-efficiency methods proposed in the literature that will help and support future researchers. Please note that the manuscript is an extended version and based on the summary of the Ph.D. thesis. This paper will give to the researchers an introduction to what they need to know and understand about the networks, WSNs, and IoT applications from scratch. Thus, the fundamental purpose of this paper is to introduce research trends and recent work on the use of IoT technology and the conclusion that has been reached as a result of undertaking the Ph.D. study.

**Keywords:** Internet of Things (IoT); IoT enabling technologies; communications; energy consumption; wireless sensor networks (WSNs); energy optimization

## 1. Introduction

The Internet is generally defined as a global system of connected computer networks that use the transmission control protocol and Internet protocol (TCP/IP) to send and

receive the data via various types of media [1]. Numerous technologies have contributed to development of the Internet in its current form [2]. This has enabled more and more devices to link together and give the opportunity for these devices/things to communicate within a local network, across different networking types, and to create a much more connected world. These devices and smart objects are becoming more and more ubiquitous in our everyday life, and have given rise to a new concept of networking which is called the Internet of Things (IoT). The history of the Internet development over the last half century [2]. It started with connecting two computers together (small network) and it is moving fast to connecting millions or even billions of physical smart devices to the internet [3].

The IoT is the inter-networking of smart objects/devices used in our daily lives that use standard communication architectures to provide new services to the end-users [4]. The IoT brings together various emerging and enabling technologies and is changing drastically what can be achieved from the Internet. The phrase "Internet of Things" was first coined by Kevin Ashton in 1999 when he used radio frequency identification (RFID) in supply chain management [5]. Since then, the IoT has been used to define a paradigm of any and all possible devices or things that can be connected and communicated to the Internet for data transfer and collection, knowledge formation and automation [4]. According to a forecast from the U.S. National Intelligence Council (NIC) in 2008 "by 2020 internet sensors may be implemented in everything such as food packets, animals, cars, plants, forests, furniture, etc.". There were 25 billion devices connected to the Internet (i.e., 3.47 connected devices per person), in the world population of 7.3 billion in 2015 [6,7]. In addition, an estimated of 50 billion connected devices around the world are deployed by 2020, which means 6.41 connected devices per person and a world population of 7.8 billion in 2020 [8,9]. Another study [10] reported the number of smart objects will reach to 80 billion in 2025 with 9.8 connected devices per person. Based on these studies.

Wireless sensor networks (WSNs) is an essential part of the IoT technology as it helps in combining heterogeneous systems, data and applications. In such systems, sensor nodes capable of detecting the required information, performing some processing and communicating with other connected nodes are the main component of these networks. However, the life of these nodes is often restricted by being powered by a battery with a limited life, constraining processing ability, memory, and radio communications [11,12]. Energy efficiency is one of the most crucial issues for WSNs. Most of the energy is consumed in data processing and transmissions [13]. This means it is not rational to waste energy on protocol overheads, the transmission of unneeded data or non-optimized transmission of data packets, especially retransmissions, due to inefficient scheduling and routing algorithms. Thus, it would be prudent to design and implement efficient load balancing schemes of energy gauge nodes to maximize the lifespan of constraint-oriented networks [14].

The paper is arranged as follows: Section 2 highlights the research motivation of the paper. Section 3 introduces an overview of IoT enabling technologies introduces in Section 3.1. A brief description of the WSNs technology that enabled the IoT revolution is provided in Section 3.2. Various sources of energy wastage and different solutions have been mentioned in the literature and are presented in Section 3.5. Section 5 identifies the gaps in the existing literature in terms of energy conservation measures that could be considered in future works. Finally, Section 6 is summarized the paper.

## 2. Motivation of the Research

The framework of the IoT is based on several enabling technologies including WSNs, cloud computing, machine learning, and peer-to-peer systems [15]. WSN is the most crucial part of the communication process of the IoT networks. In WSNs, sensor nodes capable of detecting the required information, performing some processing and communicating with other connected nodes are the main component of these networks. However, the life of these nodes is often restricted by being powered by a battery with a limited life, constraining processing ability, memory, and radio communications [16]. Energy efficiency

is one of the most crucial issues for WSNs. Most of the energy is consumed in data processing and transmissions [17]. Therefore, this paper provides a brief description of the IoT, and WSNs technology that enabled the IoT revolution. The paper also focuses on energy conservation as one of the major challenges facing IoT and WSNs, and identifies factors that affect energy consumption in such networks.

## 3. Literature Review

IoT brings an enormous number of different devices and infrastructures under the same umbrella and the consequent massive growth in connectivity of 80 billion smart devices on the Internet in 2025. Figure 1 reveals the growth of the IoT devices compared to the growth of world population [6–9,18,19]. These networks generate a large amount of data collected via these smart devices. Every two years, data doubles in size and is expected to arrive 163 Zettabytes in 2025 [20]. The volume of the IoT data will increase from 2% in 2013 up to 10% in 2021 [21]. Figure 2 reveals the growth of data from 2010 to 2025. It observes that by 2025, the volume of data will increase ten times compared to the data generated in 2016 [20,22,23].

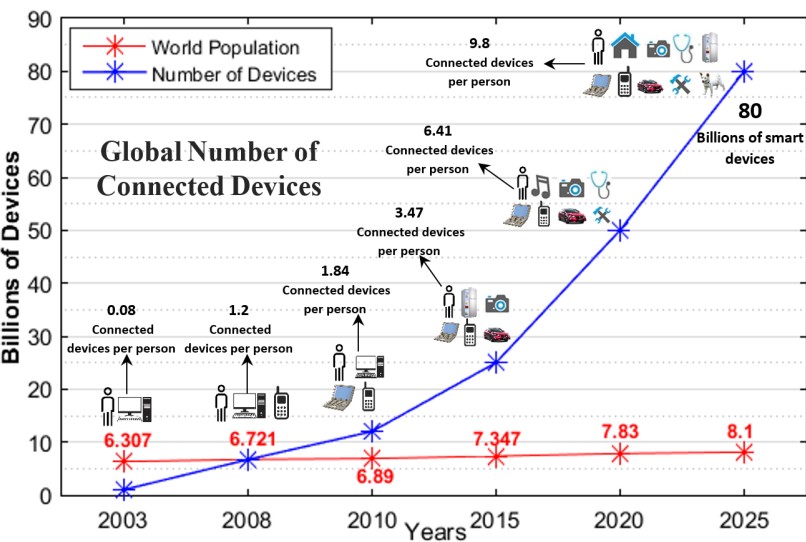

**Figure 1.** Estimated number of connected devices vs world population.

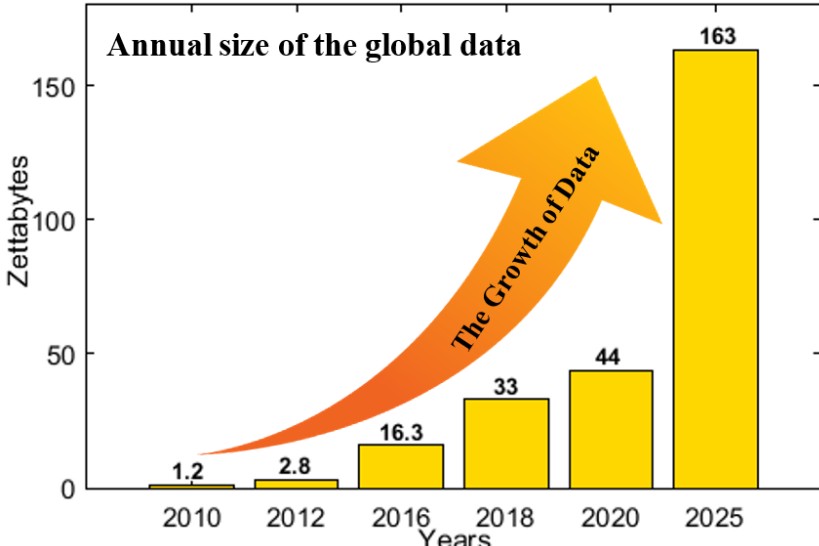

**Figure 2.** The growth of data over the years.

The concept of IoT has attracted considerable attention by governments, businesses, military, healthcare, industries and researchers [24]. It can be expanded to almost everything from refrigerators to televisions (TV), smartphones to wristwatches, home security and alarm systems, etc. [25]. For instance, smart refrigerators can tell us the end of the validity of food using RFID or which items to buy during our next shopping trip to the market. Another example, users can use their smartphones or tablets with just a single touch to control items in a house such as turning lights ON/OFF, or setting the desired temperature before arriving home, this latter is now so well developed as to be advertised by APPLE, SAMSUNG, etc. as apps for their latest phones [26].

IoT with Cloud computing both serve to increase efficiency for business and industry. The IoT generates a massive amount of data, with cloud computing providing storage and pathways for those data to travel to their destination [27]. For instance, Amazon web services are one of several IoT cloud platforms that helps people to interact with their items to be purchased through its website [28]. Additionally, the integration of IoT with medical technologies enables real-time monitoring system and data collects to improve patient health [17,29]. For example, IoT devices can be applied to track the real-time location of medical equipment such as blood pressure, heart rate, smart inhaler, oxygen pumps, wheelchairs, and other monitoring equipment. Additionally, IoT based on WSNs technology is used to monitor a wide range of sensors in the field that can detect and measure various physical phenomena such as volcanic activity, flooding, and wildfires [30]. These examples are just a few examples out of millions being implemented in IoT. Figure 3 shows some of the IoT applications in different fields.

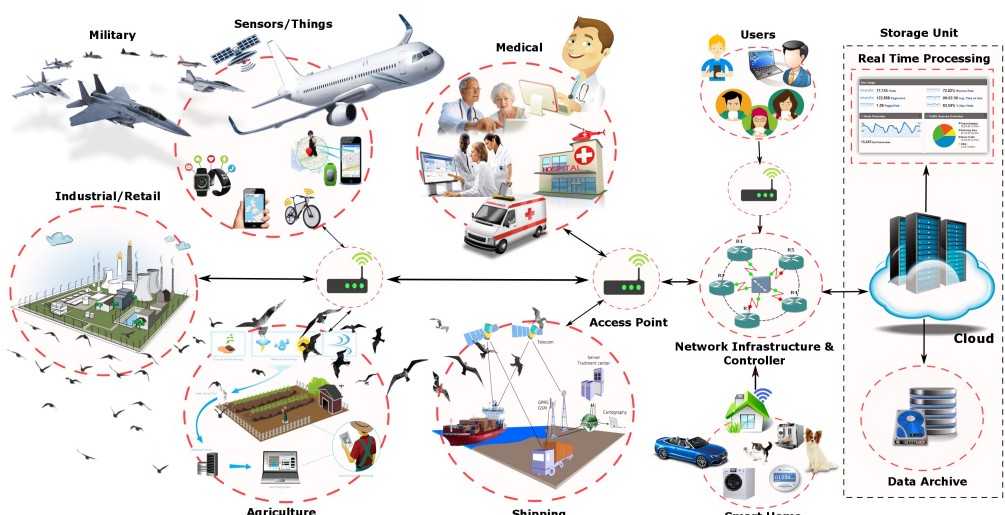

**Figure 3.** Some applications that benefit from IoT technology.

The literature review reveals that communications and IoT applications normally take the form of one of the following connections, see Figure 4:

- People to People (P2P) connection: is the data transfer/share from a user to other. For example, telephone calls, video calls and social communications. It is usually named a collaboration connection [4].
- Machine to People (M2P) connection: is the data transfer from devices such as sensor nodes, smart devices, computing devices or others to the users for analysis. For instance, weather forecasting uses smart sensors to collect the information from the sensing field and dispatches it to the remote control center for further analysis [31].
- Machine to Machine (M2M) connection: is the data transfer between devices without human interplay. For example, a car connecting and talking to another car about its lane change, congestion, accident, distance, speed, or braking intentions, etc. [32].

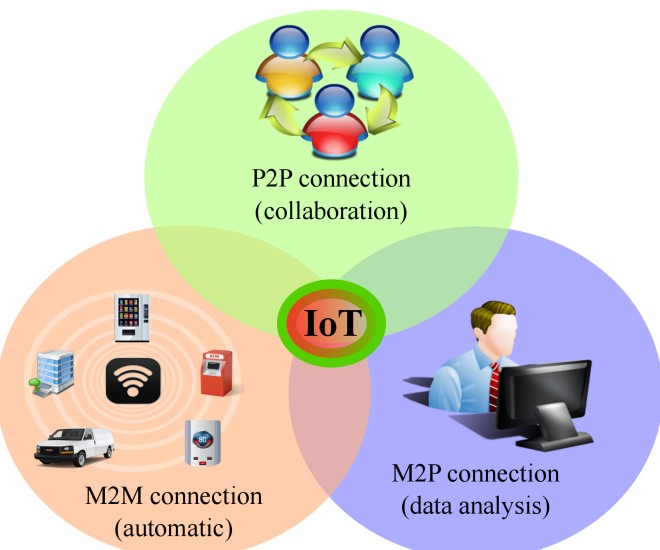

**Figure 4.** Interactions between IoT networks.

There is several studies that cover different aspects of IoT communication. For example, the survey by Atzori, et al. [33] reported the major technologies that enable the IoT evolution, both wired and wireless and the components of the wireless sensor networks (WSNs). Another study [34] presented a centralized cloud vision to enable application of IoT technology to services provision. According to [35], Internet Protocol version 6 (IPv6) is the next-generation of IP, and because it will allow for more unique TCP/IP address identifiers to be generated, it is an important innovation for future Internet communications. Thus, IPv6 will play a significant role in IoT networks.

Thus, we conclude that the main components comprising IoT technology are the human element, enabling technologies and the Internet. However, enabling technologies are at the heart of the IoT, which is possible only due to the development of technologies such as communication protocols, cloud computing, WSNs, embedded systems, mobile Internet, big data analytics, and web services [4].

### 3.1. IoT Enabling Technologies

Enabling technologies play a crucial role in realizing the IoT vision. These technologies provide connectivity, usability, capabilities, etc. that are required to facilitate efficient use of IoT applications [36]. Several studies in the literature have reported the enabling technologies and thus, this section introduces an overview of those technologies which are related to the scope of this paper.

- Cloud Computing: as the numbers of IoT smart devices increase, the amount of data generated by them also increases [37]. However, IoT devices tend to suffer from limited energy, memory, processing capabilities, etc., and their integration into the cloud is the best available way to overcome most of these issues. Cloud computing is employed to process, store, monitor and visualize the information comes from the IoT devices [38]. This means data processing and storage takes place in the cloud platform rather than on the IoT device [39], this has significant implications for IoT-constrained devices such as low-cost connectivity, scalability, interoperability, etc.
- Hardware Devices: various hardware platforms have been evolved to perform the IoT networks such as Raspberry Pi, NodeMCU (ESP8266), Arduino, BeagleBoard, FriendlyARM, etc. [24]. These devices vary from low-cost, low-power, processing units (e.g., microprocessors, microcontrollers, etc.), single-boards and software applications that can run IoT applications and communicate over the Internet [40,41].
- Wireless Communication: most IoT devices rely on low-power physical networking technologies such as RFID, Bluetooth, WiFi and IEEE standard 802.15.4 which are

essential to activate the connectivity between smart devices [42]. These technologies must be globally addressable to connect with other smart devices over the Internet, either directly or indirectly, via an IP address [43].

- Communication Protocol: IoT devices require IPv4 to connect through the Internet; however the near exhaustion of IPv4 addresses prior to the advent of the IoT and the prediction that there will be up to 50 billion Internet-connected devices by 2025 has meant that a replacement is required to permit the continued expansion of the IoT and Internet in general. IPv6 is the standard proposed to replace IPv4, and uses 128-bit addressing, allowing for a total of $3.4 \times 10^{38}$ unique addresses, instead of the 32-bit addressing used for IPv4 [44]. IPv6 has been applied to low-power wireless personal area networks via 6LoWPAN [45] which allows sensor nodes with limited resources to forward and share their data wirelessly to the other devices/things or cloud infrastructure.
- WSNs: are the most crucial part of the communication process of the IoT networks. They contain sensors embedded with a microcontroller to provide intelligence and a means of communicating via the Internet or some other network [46]. The sensors enable interaction with the physical world [47], and without the associated networks, there would be no communication between the virtual and physical worlds. The benefits of connecting the WSN to the IoT is to provide remote access and permit them to connect and disseminate the information with other devices/systems over the Internet [48].

### 3.2. An Overview of WSNs

At the core of the IoT are WSNs. It is one of the most promising wireless technology systems for enabling IoT networks. WSN contains tens of thousands of nodes connected and communicated with each other using wireless technologies [49]. Such networks involve low cost, low-range, low-power circuits and tiny sensor nodes. The main equipment of each node are a sensing and processing unit, power source, memory and receiver and transmitter unit as illustrated in Figure 5 [50].

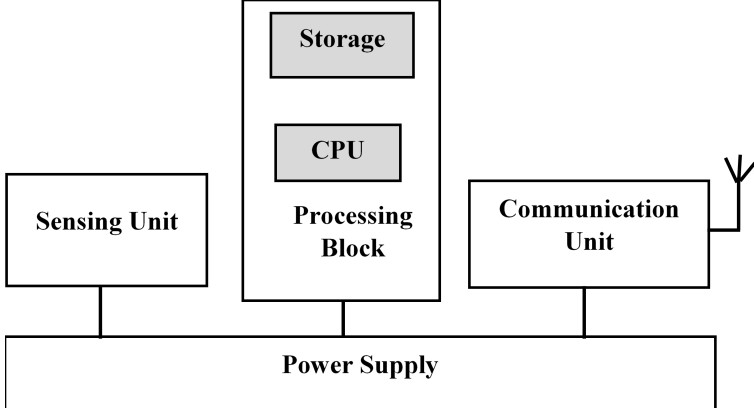

**Figure 5.** Basic components of WSN.

These units are [50]:

- Sensing Unit: is the core component of the WSN and has two functions. First, it senses information from the surrounding physical environment and converts this information into digital data. Second, it forwards the data towards the processing unit.
- Processing Unit: contains a microprocessor with a limited amount of memory. It is responsible for receiving the information from the sensing unit and forwarding the data to the transceiver after necessary processing.
- Communication Unit: combines both a radio transmitter and a receiver, and is responsible for exchanging information with other smart objects in the sensing field.

- Power Unit: is responsible for providing power to all other units. The sensor node would die, stop obtaining and/or transmitting data if the power unit stopped working. Therefore, preserving the working life of the power unit by energy conservation becomes an important and challenging issue in WSNs.

A WSN can have many types of sensors depending on the application, whether terrestrial, underwater, underground, multimedia, or mobile [51]. The main task of the deployed sensors in WSN applications is monitoring and, as stated above can include such diverse fields as meteorology [46], fire prevention [52], flood and earthquake detection and monitoring [48], mapping environmental bio-complexity and studying environmental pollution [53]. WSNs have also been used to observe the activities of animals, birds and insects [54]. Table 1 illustrates an example of WSNs applications [55]. In such applications, the smart objects are positioned over a wide geographical area and non-accessible environments.

**Table 1.** Examples of WSNs Applications.

| Type of Application | Military | Habitat | Business | Public/Industrial | Health | Environment |
|---|---|---|---|---|---|---|
| Tracking | Enemy Tracking | Animal Tracking | Human Tracking | Traffic Tracking | Patient Tracking | Tornado Tracking |
| Monitoring | Security Detection | Animal Monitoring | Inventory Monitoring | Machine Monitoring | Patient Monitoring | Weather Monitoring |

*3.3. WSN Communication Architecture*

A WSN is similar to a wireless ad hoc network, since both are self-organized and multihop networks [16,56]. WSN is used to monitor and record specific phenomena and cooperatively pass data wirelessly through a gateway (base station/sink) to a central location as shown in Figure 6. The more modern WSNs are bi-directional (two-way communication), thus enabling control of the activity of the sensors [57].

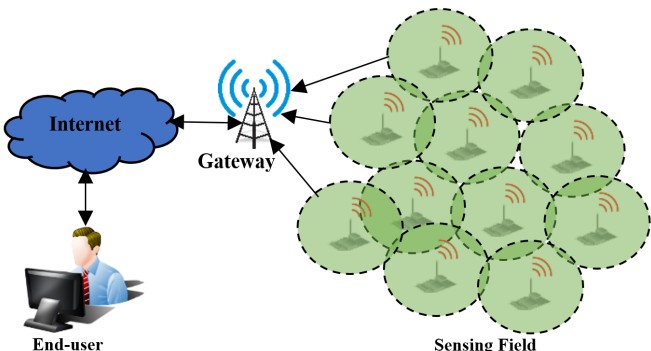

**Figure 6.** WSN communication architecture.

In large-scale networks, the sensory network is often partitioned into subgroups, with each group having both sensor nodes and a single cluster-head (CH) node. This clustering approach has several advantages for sensor networks in terms of energy consumption, delay, network communication, etc. [58].

Sensor nodes establish connection with each other wirelessly and individually collect data from the surrounding environment, perform simple computation processes and then forward the information to its associated CH node via a single hop or intermediate nodes [59]. The intermediate node serves as a data sender and path for other sensor nodes towards the CH node. These nodes make forwarding decisions (i.e., routing) based on their knowledge of the network [60]. The CH nodes can be elected randomly or based on one or more criteria such as number of neighboring nodes, transmission distance to the final destination, residual energy, where the BS is the master node which gathers the data

from all sensor nodes and processes it, and then disseminates these data to the intended destination [61]. The task of the CH node is to gather the information from its member nodes, compress it and then disseminate it to the base station. Most of the literature on WSNs is related to a search for proper clustering, optimal path, and aggregation methods that can significantly minimize the energy depletion and lengthen the network lifetime. Figure 7 reveals a WSN communication architecture which splits the sensing field into two groups and each group has several nodes that are linked with each other to main CH node and these CH nodes are connected to the BS.

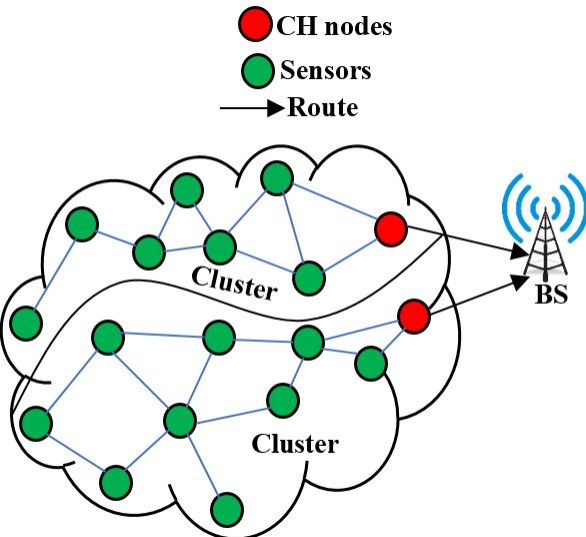

**Figure 7.** A typical clustered WSN.

Load balancing technique is an important aspect to maximize the network lifetime of the sensors by minimizing energy consumption. According to an investigation by [62], authors use a new cluster formation scheme in the sensing field to manage the load balancing issue in the deployed area. The CH nodes in the proposed scheme are selected dynamically for a fixed interval. Although the static route is created in the designated clusters by applying the AODV routing algorithm. The scheme balanced the energy consumption of the nodes and thus extended the network lifetime. In another study [63], authors introduce a novel routing protocol called content-based adaptive and dynamic scheduling (CADS). The protocol is used two ways communication model for the WSNs. The CADS is avoided redundant data and thus reduced the forwarding of unnecessary data packets, which as a result, prolongs the network lifetime.

*3.4. IoT-Based WSNs*

The integration between WSNs and the IoT has a crucial role to play in many applications and facilitates the universal accessibility of data, and close-to-real-time decision-making [64]. The sensor nodes connect to the Internet dynamically to cooperate and achieve their tasks; however, most of the connected sensors are constrained within their ecosystems which have limited memories, processors and power sources [65]. When a WSN is integrated into the Internet as part of the IoT, numerous decisions are required regarding that integration, including; mode of communication, hardware, computational cost, security, big data, and battery power [66,67], see Figure 8. All these issues must be addressed to achieve the full advantages and benefits of such integration, but energy depletion is considered to be one of the more important aspect. This is due to the crucial role that these sensors play in determining the lifetime of the entire network.

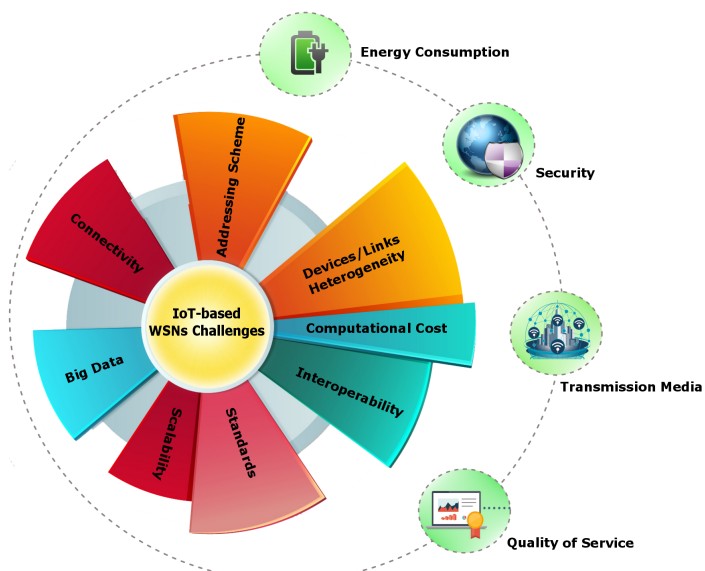

**Figure 8.** Necessary considerations for IoT-based WSNs.

### 3.5. Reasons and Solutions for Energy Consumption

As mentioned above, energy consumption, in general, is one of the most challenges and problems in IoT-based WSNs applications [64]. Sensor nodes are often operated by batteries and thus, these nodes can only run for a limited period of time. There are several reasons that could lead to exhaust the limited battery of the sensor nodes. However, several attempts have been made to minimize the energy usage of the smart objects and thus lengthening the lifetime of the network. Network lifetime is one of the most crucial metrics for the evaluation of sensor nodes but, in the literature, there are different definitions of network lifetime. Generally, it is realized as the length of time that sensor nodes would be fully operational. In other words, network lifetime is the time until the first node dies [68]. Another definition of network lifetime is the time until the first node or group of nodes in the sensing field exhausts runs its energy [69]. A node can only fulfill its mission as long as it is live, so losing a node would damage the network which would lose some of its functionalities. Hence, the main aim of any energy-efficient protocol is to keep the nodes alive for longer and thus prolong the network lifetime.

Various sources of energy wastage and different solutions have been mentioned in the literature and are demonstrated in the following sub-sections.

### 3.6. Sources of Energy Wastage

Several studies have revealed that the communication unit is relatively greedy for energy [68]. In WSNs, most of the energy is wasted in the processing, receiving, or transmitting of data to fulfill the requirements of the application [70]. It is clear that reducing data transmissions will economize the energy of these smart objects [68]. Regarding communication, several studies have found that a great amount of energy is dissipated in ways that make no useful contribution to the application, such as [71]:

- Collision: when two or more packets reach the sensor node at the same time and thus a packet collision occurs [72]. Thus, the packets are either discarded or sent back to their originating node, then retransmission of these packets is needed which rises packet latency and energy depletion which adversely affects the network lifetime [73].
- Overhearing: is a significant waste of energy, especially when node density is high and traffic load is heavy. When a node sends a packet, all sensor nodes in the network located within its transmission range distance receive the packet even if these nodes are not the proposed destination [74,75], see Figure 9. Node A wants to deliver its information to Node B. However, many surrounding nodes are within radio range of

Node A. All these nodes will receive the data from Node A. Energy is consumed when a sensor node transmits or obtains the data that are intended for other nodes [75]. Please note that Node A will also receive data from its surrounding nodes when they transmit their data.

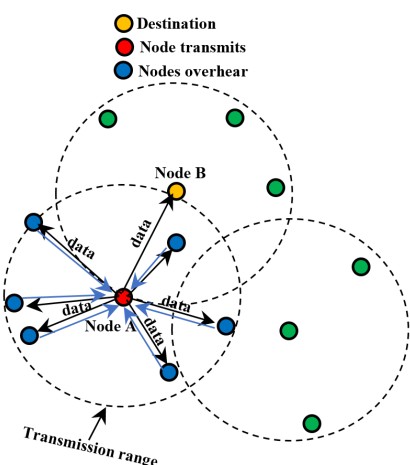

**Figure 9.** A source node transmits to its destination and neighboring nodes overhearing the communication.

- Control packet overhead: is a combination of excess memory, bandwidth, computation time or other resources to execute a specific job. Thus, it is crucial to process the minimum number of control packets that enable the transmission [71].
- Idle listening: happens when a sensor node must stay open to an idle channel to receive possible traffic [71], thus a sensor surrounds with many neighbor nodes will be active most of the time. This is due to overhearing transmissions, neighbor nodes discovery [76] or a node may use numerous paths to deliver data to a neighbor nodes [77]. Obviously, a node with less idle listening time has better energy retention than other nodes [71].
- Interference: each node with two or more nodes within transmission range suffers from interference generated by the surrounding nodes. Interference increases with increase in the number of neighboring nodes [78]. It increases both congestion and conflicting transmissions, and then retransmission may happen. Therefore, avoiding higher node interference could reduce packet loss and thus minimize the overall energy wasted of the network [79].
- Redundant Data: nodes are generally deployed randomly which can mean that there are some regions monitored by two or more sensors at the same time [80]. However, this type of deployment will increase the reporting of redundant data in the network. As a result, energy is wasted aggregating, processing and transmitting redundant data [13]. Energy consumption could be minimized by avoiding the unnecessary operation of a node.
- Distance: the transmission distance ($T_d$) between nodes is a very important aspect of energy efficiency. The communication between a node and its associated CH node and the intended destination can be either single or multiple hops. Since energy consumption for transmission is proportional to the square of the distance (see Equation (1)) [81], so the power required for transmission increases rapidly with distance, which means single-hop transmission maximizes energy depletion if the size of the network is large.

$$E_{Tx} = k(E_{elec} + \epsilon_{amp} * d^2) \tag{1}$$

where $E_{Tx}$ is the energy used to dispatch a chunk of data ($k$) from the node to the next-hop node. $d$ is the distance between the source node and next-hop node. $E_{elec}$ is illustrated the energy dissipated to perform the transmitter/receiver board, and $\epsilon_{amp}$

is the energy spent in transmission process to amplify the signal enough to reach the next target.

Thus, most of the literature shows that multihop communication is the best way to minimize the transmission distance between nodes. Figure 10 shows single and multihop scenarios between nodes. A lower transmission distance between a node and next-hop target/CH/BS reduces energy depletion of a node and prolongs the network lifetime [68].

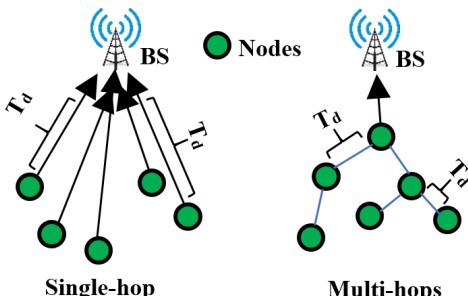

**Figure 10.** Single-hop and multihop scenarios.

- Non-Clustering: direct transmission distance from a source to the next-hop node can reduce the sensor network lifetime significantly due to the additional energy consumption. As a solution, hierarchical routing protocols are adopted, see Figure 11 which shows chain-based, tree-based and cluster-based protocols, which are the most commonly used protocols [82]. In a chain-based method, sensor nodes are organized chain-like where one of these nodes is elected to serve as the CH node to transmit the information coming from all sensors to the BS [83]. With cluster-based, the sensing field is partitioned into subgroups and each sub-group has some sensor nodes connected to a CH node to forward their information to the BS [84]. In tree-based clusters, the collected data are forwarded from node to their associated CH node based on multihop concept [85]. For sensor networks, clustering is the best solution for reducing communication costs and maximizing network lifetime.

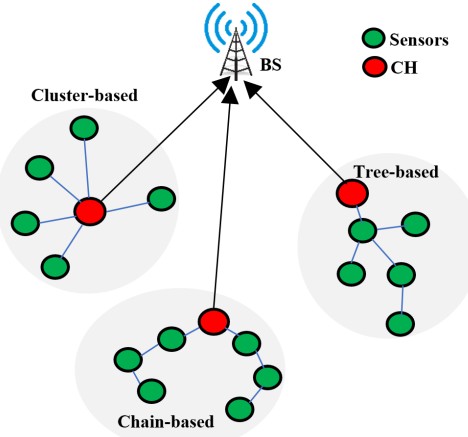

**Figure 11.** Hierarchical clustering in WSN.

*3.7. Taxonomy of Energy Consumption Solutions*

There are many methods available in the literature that can effectively minimize energy usage and lengthen the network lifetime of WSNs and IoT networks such as path selection [86], scheduling data [65], an efficient data aggregation [87], etc. These methods can broadly be classified into several categories as summarized in Figure 12 which presents a taxonomy of energy consumption solutions and techniques presented in the literature. These solutions and techniques are explained in the following sub-sections.

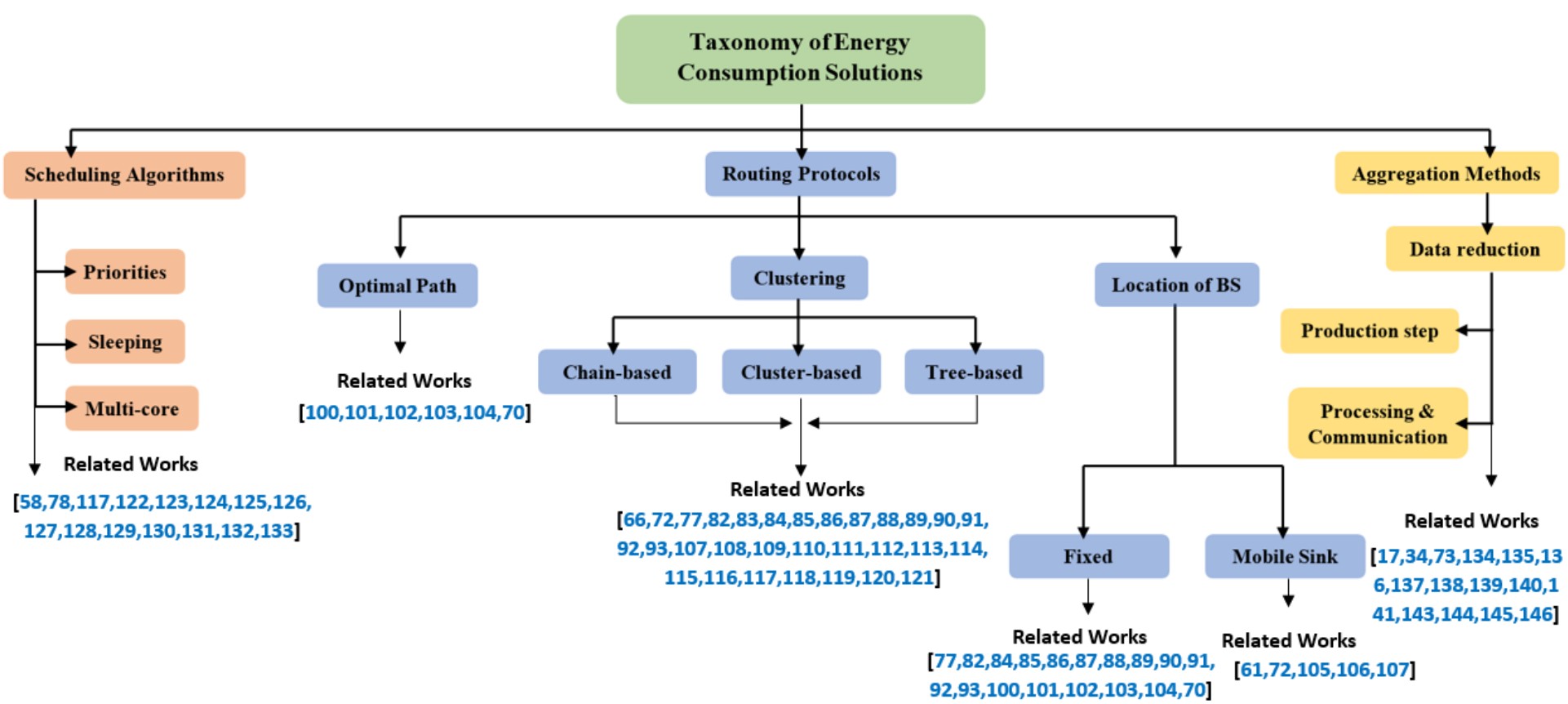

**Figure 12.** Taxonomy of energy consumption solutions.

### 3.7.1. Routing Protocols

One of the most significant current discussions of IoT based on WSNs, is the generation of an unprecedented amount of data [37], and how to select optimal paths for transmitting such vast quantities to the final destination. Energy-efficient routing algorithms are implemented to decrease the energy depletion and lengthen the network lifetime for these sensor networks. Several routing algorithms have been investigated by various authors to conserve energy. Most involve investigating the concept of using clustering the sensing field and CH nodes selection, electing the optimal path from the source node to the required destination, or manipulating the location of the BS [88].

(i) Cluster-Head Node Selection

Various strategies are used in the literature for CH nodes election process to optimize energy usage. The most common three are: low energy adaptive clustering hierarchy (LEACH) [84], hybrid, energy-efficient and distributed protocol (HEED) [89] and power-efficient gathering in sensor information systems (PEGASIS) [90]. We present a brief survey of LEACH, HEED and PEGASIS in which nodes are partitioned in many forms for data collection and communication protocols.

(a) **Low Energy Adaptive Clustering Hierarchy (LEACH)**

LEACH is one of the most interesting strategies, in which the CH node is elected based on a probabilistic approach and the amount of energy remained of the CH and the system is rotated at different time intervals [84]. A sensor node that has already been the CH cannot be elected again for some rounds. The selected CH node broadcasts to the network and creates a schedule for each node in its cluster to send its data. Each node connects to the CH with a single hop and chooses a random number between 0 and 1, then compares the number with a threshold value $T(n)$. A node becomes a CH in each round if the random number is less than the following threshold:

$$T(n) = \begin{cases} \frac{1}{1-P(r \mod 1/p)} & \text{if } n \in G \\ 0 & \text{if } n \notin G \end{cases} \qquad (2)$$

where $G$ is a group of sensor nodes that have not been picked as CH node in the previous $1/p$ rounds. $r$ defines the most recent round, $P$ is the required percentage of CH. The CH node collects the information from all sensors connected to it, compresses the information and then forwards it to the ultimate receiver. Every node will be in standby mode except when sending to its CH. Figure 13 shows a cluster organization for the LEACH protocol.

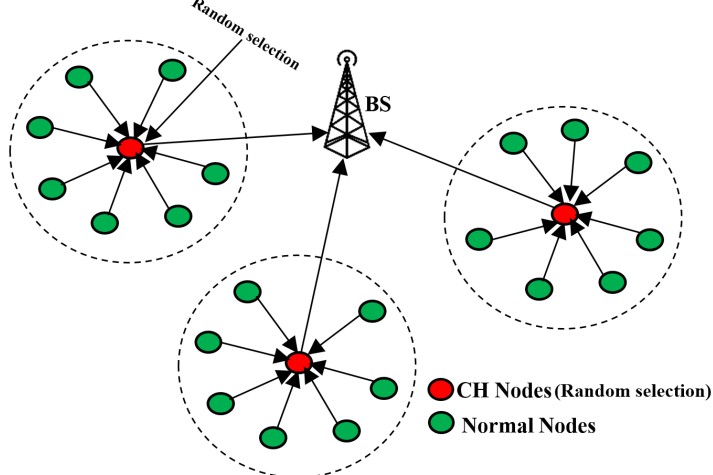

**Figure 13.** Example of LEACH protocol architecture.

Several studies have been published on modifications of the LEACH protocol, such as LEACH-C and energy-balanced LEACH [91,92].These studies tried to overcome the problems associated with LEACH (i.e., random process selection of CHs) and further minimize the total energy consumption for WSNs.

In the LEACH-C scheme, each node in the sensing field can calculate its energy level and send the information about its location (possibly using GPS) to the intended destination. The intended destination uses a centralized clustering algorithm to select the CH nodes. Once the clusters and related CH nodes are computed, then the BS chooses a node with more energy and broadcasts a packet to all sensor nodes that consist of the ID of each CH node. If the ID matches, then a node is the selected CH node and its intended destination is the BS. Otherwise, a node must gather and forward the information to the CH node.

LEACH-C provided better clustering and longer lifetime than the LEACH protocol. However, energy-balanced LEACH (E-LEACH) enhances the CH node election by considering the remaining energy of each node. Initially, each node has the same residual energy, and the CH nodes are elected randomly. From second round, each sensor node with the highest remaining energy will become the CH node of its cluster. The E-LEACH protocol uses master cluster heads (MCH) to relay packets for those CH nodes that are away from the required destination.

Similarly, Arya, et al. [93] introduced a modification of the LEACH protocol named the energy aware multihop multipath hierarchy protocol (EAMMH). This approach introduced a new routing strategy and clustering formation to transfer the data. The proposed method divides the sensing area into subgroups and each group has number of child nodes and main CH node. The main CH should be an optimum distance from these child-CH nodes. This means the distance between the CH and its member nodes should be balanced to minimize energy consumption and therefore increase the lifetime of network. The EAMMH scheme outperformed LEACH in terms of energy preservation by 23% but the main CH nodes can be overloaded and quickly drained of energy when surrounded by many child-CH nodes.

Cengin et al. [94] proposed the energy aware multihop routing (EAMR) method for WSNs. The EAMR proposes fixed clusters to provide communication between the sensor nodes and the BS. In this protocol, when a sensor node is attached to a cluster, it will be a member for that cluster for the whole network lifetime. The selection of CH nodes is repeated each round, the proposed protocol allows a sensor node to act as a CH node until its energy falls below a threshold value. Sensor nodes located close to the BS forward their data direct to the BS. However, the remaining CH nodes forward their packets to the BS through intermediate nodes. The EAMR extends the network lifetime by achieving steady clusters and reducing the number of CH node changes.

Although the LEACH and its derivative protocols paved the way for implementing energy-efficient routing protocols, they all suffer from one fundamental problem. A node uses single-hop routing within clusters thus, it is not suitable to sensor networks for large geographic area. Additionally, a node that is elected to be CH will die quickly if a larger area is to be supported. Because some CH nodes are positioned far away from the final destination, the resulting large transmission distances lead to large energy consumption.

(b)     Hybrid, Energy-Efficient and Distributed Protocol (HEED)

HEED is the other common method of CH node selection. The proposed protocol overcomes the drawback of LEACH by achieving equal and uniform distribution of CH nodes in the sensing field. In this approach, the CH node selection is based on the residual energy of each node and node proximity

to its neighbors or node degree (minimum communication cost) [89]. HEED defined the average of lower energy levels (AMRP) required by all $M$ sensor nodes within the cluster range, to reach the CH node as:

$$AMRP = \frac{\sum_{i=1}^{M} MinPwr_i}{M} \tag{3}$$

where $MinPwr_i$ is the lower energy level desired by node $i$ to communicate with the CH. Each node is assigned to only one cluster, and the node independently makes its decision based on local information to join a CH node via a single hop. Based on Equation (4), in HEED, every sensor hub sets the likelihood $CH_{prob}$ of turning into a CH as:

$$CH_{prob} = C_{prob} \times \frac{E_{residual}}{E_{max}} \tag{4}$$

where $E_{max}$ is the total energy of the node and $E_{residual}$ is the evaluated remaining energy in the node, which is typically similar for all nodes. $C_{prob}$ is only used to limit the initial CH announcements, and has no direct impact on the final clusters.

A CH node is either a temporary CH, if its $CH_{prob}$ is < 1, or a last CH, if its $CH_{prob}$ has achieved 1. Analysis of the relative performance of HEED and LEACH showed that HEED improved the network lifetime by 10% [95]. Figure 14 introduces an example of a network topology implemented by the HEED protocol.

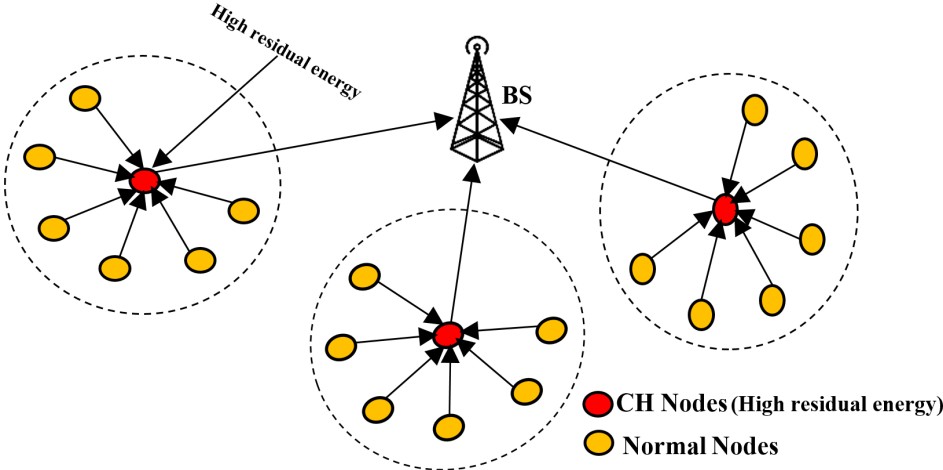

**Figure 14.** HEED protocol architecture.

Several researchers have attempted to overcome the limitations of HEED protocol (such as more CHs are generated, the locations of the CHs, etc.) and improve its performance [96]. One example is the heterogeneous hybrid energy-efficient distributed (H-HEED) algorithm. This algorithm divides the sensing field into clusters and each cluster has some sensor nodes. The H-HEED protocol finds the center of each cluster and then allocates the node nearest the cluster center. The H-HEED protocol re-computes the cluster centers with a new assignment of nodes and allocates a node to clusters until clusters do not change for a given number of iterations. However, in this protocol, several iterations are performed to form the clusters and select a CH, this is an overhead that consumes a significant amount of energy [97]. Nevertheless, the proposed scheme increased the network lifetime of the sensor nodes by 63% [98]. Another study [99] proposed an energy-based rotated

HEED (ER-HEED) protocol for WSNs. Here, the clustering formation and CH node selection are implemented based on the HEED protocol. Therefore, the selection of CH node among sensor nodes in each cluster is based on the node with the highest level of energy. ER-HEED improves the HEED protocol by reducing the HEED cluster selection to minimize energy wasted and lengthen network lifetime.

In [100], a new multihop routing strategy was proposed, the cluster heads enhanced hybrid, energy-efficient distributed HEED method (E-HEED) for WSNs. The E-HEED chooses the CH node according to the HEED protocol, and then grades the CH nodes according to the least transmission distance from the BS. It was claimed that the E-HEED protocol lengthened network life by 0.8 % compared to HEED.

(c)     **Power-Efficient Gathering in Sensor Information Systems (PEGASIS)**
PEGASIS is another CH node selection technique. This approach is to form a chain among the sensor nodes for the transmissions, see Figure 15 for the architecture of the PEGASIS routing protocol [90]. Each node receives the data from one neighbor node and transmits it to another. Two nodes at the end of the chain forming the routing structure will forward the information through the other nodes to the single leader node (CH node) and then the CH sends these data to the intended destination. The CH node is randomly elected to transmit the gathered data to the intended destination. PEGASIS is aimed to minimize the transmission distances between sensor nodes in the sensing field, and thus the energy depletion of each sensor is minimized. However, only one node is picked as a CH node per round. It this may become a bottleneck that causes delay and retransmission of some of packets. It also increases the rate of packet transmission on the node selected as a leader and thus depletes its energy quickly.

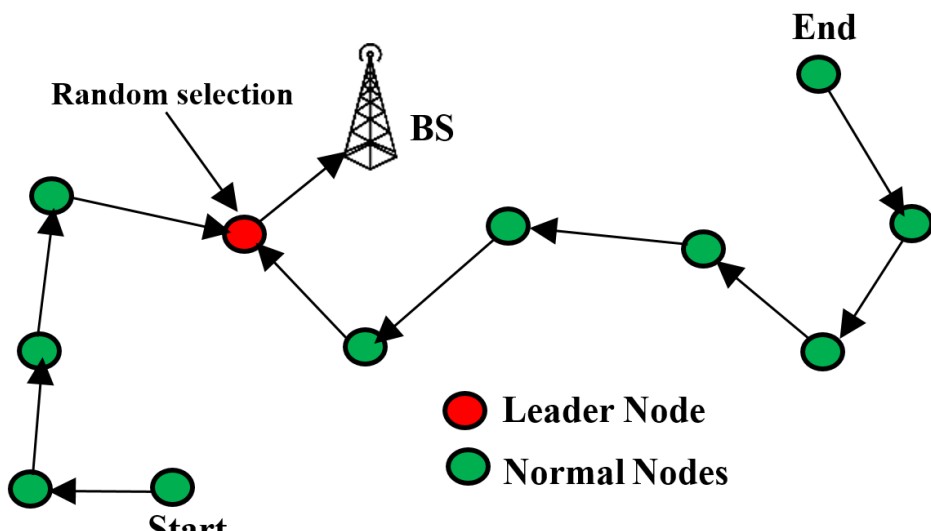

**Figure 15.** PEGASIS protocol architecture.

Table 2 presents characteristics and comparisons of LEACH, HEED and PEGASIS based on the more important metrics:

**Table 2.** Comparison and classification of some of routing methods in WSN.

| Parameters | LEACH | HEED | PEGASIS | References |
|---|---|---|---|---|
| Type of protocol | Hierarchical | Hierarchical | Hierarchical | [89,101] |
| Data delivery model | Cluster-based | Cluster-based | Chain-based | [102] |
| Nodes distributed | Random | Random | Random | [103] |
| Node mobility | Fixed | Fixed | Fixed | [102] |
| Multihop | No | Yes | No | [102] |
| Clustering Method | Distributed | Distributed | Centralized | [103] |
| CH selection | Threshold | Residual Energy | Threshold | [83,84,89] |
| Relay node | CH | CH and nodes | nodes | [83,84,89] |
| Data aggregation | Yes | Yes | No | [102] |
| Scalability | Low | Moderate | Low | [104] |

More recently, authors [105] investigated, a new routing technique called destination-oriented routing scheme for energy-balanced WSNs (DORA). The DORA aims a new multichain routing method to transmit the data to balance energy for the sensor nodes. In this protocol, the optimal transmission distance between any two nodes in the sensing field is derived by the mathematical analysis model. The proposed protocol reduces energy consumption for the nodes and thus extends the global network lifespan. However, in this protocol, any node in the sensing filed might be connected with two or more multichain based on the transmission range of the node, and thus a node may be sends same data through two or more paths to the final destination. Recently, ARUN et al. [55] gave a comprehensive review of the IoT and WSN technologies for medium access control (MAC) protocols. The review focuses on the MAC layer protocols and common causes of energy consumptions. Early studies by [106] have investigated a new routing technique LLND protocol which is defined the MAC behavior for IoT networks that run inaccessible environments. Sensor nodes interconnected by wireless links with dynamic and lossy wireless link conditions, resulting from interference, channel fading, or heat/dust/ moisture physical environment, are classified by Low-power and Lossy Network (LLNs). An Adaptive Scheduling MAC (AS-MAC) method was carried out by [107]. The proposed protocol is aimed to make nodes to decide that stays active or sleep depending on traffic load. Therefore, if the traffic load is high, AS-MAC can achieve rapid data dissemination and reduce transmission latency by scheduling more transmission. However, if there is a smaller amount of the traffic load, sensor nodes switch to a sleeping mode in a timely manner, such that idle listening is mitigated, and energy conservation is achieved.

(ii) Optimal Path Selection

Several studies have considered optimal route selection for energy-saving in WSNs. The shortest route approach is a commonly used methods for constructing routing trees in the many-to-one WSN [108]. The potential advantages of shortest path are lowest energy consumption and minimum time delay. Banerjee, et al. [109] investigated a heuristic algorithm based on multi-hops that perform geographical routing. This protocol selects a route with the fewer hops and distance from the source node to the target. The proposed scheme reduces the end-to-end node delay. In [110], authors introduced a distributed shortest path routing network from a source node to the ultimate receiver. The resulting algorithm provides best link cost and maximum network lifetime.

Cota-Ruiz, et al. [111] demonstrated a new routing technique that can calculate the distance between two non-neighbor nodes in multihop WSNs. This method finds all possible routes between a source node and the ultimate receiver with the fewer hops. This leads to minimizing the energy depletion and delay of the network overall. Another study [112] proposed a new centralized energy-efficient clustering algorithm for WSNs. This is the distance energy evaluated (DEE) protocol which selects the CH nodes according to the ratio between remaining energy of a node and distance. The

probability of being CH is determined according to the node's initial and residual energy. The DEE protocol extends the network lifetime by reducing unnecessary traffic.

Most studies have not considered the shortest path combined with balancing the load traffic in each node along the path to deliver data. A node that is surrounded with many neighbor nodes (within transmission range) has less energy due to overhearing, neighbor nodes discovery, or a node may be used for many paths to deliver neighbor nodes' data [77].

(iii) Manipulating the Location of Base Station

Several studies have proposed manipulation of BS/sink location as a means of reducing energy depletion. They found that the network lifetime of the network can be extended by reducing the transmission distances between sensor nodes. In the work of Grossglauser, et al. [113], the idea of a mobile sink (MS) was proposed, where the sink moves in a prescribed path to gather the information in the sensing field. In such a protocol, all nodes regardless of distance will establish a direct connection with the sink. Therefore, the total link length of the network will be very high, especially when a node is located on the border of the network, consuming more energy than other nodes which are close to the sink. The optimal location for a mobile sink (OLMS) for WSNs is suggested by [114]. In this approach, clustering is achieved, and CH nodes are elected at each round. The proposed protocol determines the best location of the MS based on the minimum energy cost for data delivery of CHs and thus reduces energy depletion and lengthens the network lifetime.

In [68], the authors also examined a tree-based mobile sink (TBMS) technique. The proposed study implements a sorting algorithm and the multihop technique to generate the routing structure. The proposed method introduces a MS that gathers the data from the sensing field but in a way that reduces the hop distances and thus elongates the network lifetime of the network. However, authors assume that the MS moves randomly in the sensing area. Therefore, there is no guarantee that the MS will cover all the sensing area, or it might take too long when the sensing field are extended. Of course, if the speed of at which MS moves is too slow or fast, then it can cause more delay and high packets loss.

However, some important factors such as interference effects and dynamic network topology should be considered when designing WSNs routing protocols. This is because of the challenges that may arise as a result of the characteristics of the environment in which these networks are deployed [79,115].

- Interference Effect

  High node density in the sensing field, can lead to interference effects which can adversely affect energy consumption in sensor networks. According to an investigation by [79], interference occurs during transmission and can cause packet loss. In such a case, lost packets need to be retransmitted and every retransmission is energy wasted [73]. Thus, these authors suggested avoiding paths with higher interference levels [116]. In [117], the authors proposed a new routing strategy that chooses a path with less interference of transmitted data. The proposed method balances the traffic load and significantly reduces congestion in the network. An energy aware interference sensitive geographic routing (EIGR) was investigated by [118]. The EIGR adaptively uses an anchor list to guide data delivery and chooses the less interference route from the energy optimal relay region for data delivery. The EIGR adjusts the transmission power which is only required to disseminate the information to the forwarding node. The proposed protocol focuses on reducing interference and minimizing the total energy depletion of the network.

  Other researchers [119] have addressed the problem of interference in WSNs, and here the proposed scheme detects the shortest path from source node to the ultimate receiver which avoids interference areas based on an ad hoc, on-demand distance vector (AODV) protocol. Liu, et al. [120] introduced a full-duplex BackCom network,

where a novel time-hopping spread-spectrum (TH-SS)-based multiple-access scheme was implemented. The proposed protocol enabled simultaneous forward/backward information transfer from one device to another. The interference in such networks is suppressed by the proposed multiple-access scheme based on the TH-SS technique and allows wireless energy harvesting from interference.

However, these strategies did not consider the interference caused by neighboring nodes of the next-hop node. Increasing the surrounding neighbor nodes adjacent to each node (within transmission range) generates an increase in interference [121]. As a result, increasing the packets loss and decreasing the network lifetime.

- Dynamic Network Topology

  With multi-hopping, sensor nodes depend on intermediate nodes in the network to disseminate their packets to the final destination. Some of these intermediate nodes may be failed or blocked due to exposure to physical damage, interference, harsh environment or lack of power during transmitting and receiving packets [122]. The probability of node failure rises with the increases in the sensing field and number of sensor nodes. A node is announced as a failure node when a sensor cannot send/transmit packets with its neighbor nodes for more than a specific period of time and thus eliminated from the routing path. Such node failure should not affect the overall sensor network [123]. WSN routing methods should be able to recover from the failure of a sensor node [115]. Therefore, a routing protocol must pick and connect with new sensor nodes (within the range of transmission) dynamically to forward the data gathered by other nodes to the final target. For example, Figure 16 clearly reveals that source1 forwards its data to the final target via some intermediate nodes. Unfortunately, *path*1 and 2 failed to pass the source1 data to the ultimate receiver due to failure of some nodes. Hence, a new path is required to disseminate the packets to the final destination (i.e., *path*3).

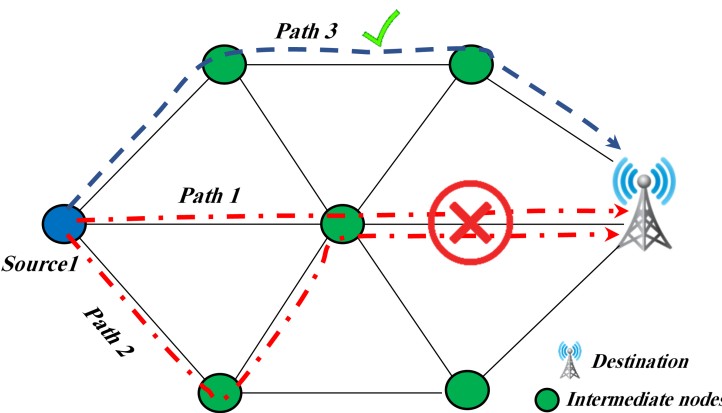

**Figure 16.** Recovery from node failure. *Paths*1 and 2 failed to carry the data from source1 to the ultimate receiver. Therefore, source1 establishes another path (*path*3) to deliver its packets.

Several studies have been carried out to provide routing protocols that help to recover from a failed node in the network. Most of this research focused on providing a backup node or finding an alternative node to avoid link failure from source to destination. According to an investigation by [124], the low-power wide area network (LPWAN) is one of the best and promising solution for long range communication and low power consumption for IoT and M2M communication applications. In different study [122], authors reported a mobile sensor node acting in cooperation with a static node to fill gaps created by faulty nodes in the sensing field, which resulted in overcoming the failure issue and increasing the network lifetime. Other work [125] proposed a new procedure that could replace the dead CH node with backup cluster heads (BCH) in the case of CH node failure. One study [126,127] has proposed an energy-efficient backup and recovery node selection for IoT networks. The system includes

backup nodes which are in sleep mode until required due to a node failing and then are enlivened. This results in energy-efficient solution and maximizes the lifetime of network.

In another study [128], the authors proposed a new algorithm to create primary and alternative paths in the network. The proposed method reroutes the traffic from nodes connected directly to the failure node, and reroutes the traffic in an alternative path. In addition to [128,129] suggested a novel path redundancy-based algorithm which called for dual separate paths (DSP). The DSP algorithm provides fault-tolerant communication for WSN applications. This protocol implements two separate paths between a node and the intended destination and thus improves the network traffic performance. The cluster-head recovery algorithm (CHRA) [130] uses a check-pointing techniques to create a recovery route for each node and in each cluster . In the case of CH failed, a recovery route is established for sensor nodes connected to the failed CH node. In the wireless ad hoc networks, it is decentralized to route the packets from the source to target. It also does not need any particular infrastructure such as backbone, access points, etc. Ad hoc On-Demand Distance Vector (AODV) routing is being one of the standard protocols in wireless ad hoc networks. It uses when two or more endpoints do not have a valid active route to communicate each other [131]. In another study [132], dynamic source routing (DSR) is a routing protocol for wireless mesh networks. It is similar to AODV protocol where it forms a route on demand when transmitting packets. However, it uses source routing instead of relying on the routing table at each intermediate node.

Recent study by [133] revealed a new saving routing mechanism, named energy-efficient cooperative Scheme for Heterogeneous WSNs (EERH). The EERH scheme is dynamically established the routing paths according to the transmission directions of event packets and the residual energy of the underlying sensors and their neighbors. According to [134] presented energy and collision aware WSN routing method for IoT networks. The proposed algorithm is based on AODV protocol; however, it replaces the hop count metric with the link quality and collision count. The protocol improves performance in terms of path stability, energy efficiency, network lifetime and delay. Study by [135] proposed a gateway clustering energy-efficient centroid (GCEEC)-based routing protocol. In this protocol, the sensing filed is divided into clusters and each cluster has a gateway node and the CH node in this cluster is chosen from the centroid position. The gateway reduces the data traffic from the CH node and dispatches the data to the final destination and thus extend the network lifetime.

### 3.7.2. Scheduling Algorithms

Sensor nodes around a CH node naturally create a many-to-one traffic pattern [85,136]. Congestion generally happens when the traffic load on a particular node exceeds the available buffer capacity which leads to successfully delivery for only some of the packets and thus packet retransmission is required [137]. Retransmission of data will, of course, consume additional energy. Most previous research does not consider packet overheads due to the retransmission of packets. For instance, when a connection-oriented protocol such as the transmission control protocol (TCP) [138] is established, for reliability it uses three-way handshakes to establish the connection between the source and destination. This leads to a significant increase in network traffic and thus increases the data transmission rate and volume. Therefore, during the implementation of the scheduling techniques, packet overhead must be considered since retransmission adds a burden on the network and reduces the lifetime of network. To minimize power and memory usage, superior scheduling protocols that consider packet overhead are required.

Several scheduling methods have been investigated for use in sensor networks. A scheduling algorithm for nodes that are positioned between two or more clusters (border nodes) is introduced in [65]. A switching technique between listening and sleeping modes is adopted by the nodes in this scheduling approach. However, a node that is

placed between more than two clusters, will often switch to the listening mode and this will cause to the unnecessary of energy wasted which decreases the network's lifetime. The proposed protocol introduces a unified scheduling method to solve the problem of diversified scheduling of border nodes.

Another study by Gupta and Rao [139] proposed demand-based coverage and a connectivity routing protocol to provide the desired coverage and meet connectivity requirements in WSNs. The idea was to use a probabilistic approach to calculate and reduce the sensing range of the sensors. It also uses a sleep scheduling protocol to switch ON/OFF the communication radio which results in saving energy. Abdullah and Yang [126] proposed clustering IoT networks into sub-clusters and placing within each cluster, a broker (CH node). The broker was deployed to collect data from the nodes around it and send the data to the ultimate receiver. The short process time (SPT) method was performed at the broker level to elect and forward data based on their arrival time. Each message is demonstrated as Mess($T_{trans}$, $R_{time}$), where $R_{time}$, $T_{trans}$ are successful transmission time and request time periods, respectively. The SPT algorithm was used when the network was unstable (traffic intensity > 1). The SPT algorithm promoted IoT system efficiency by enhancing service response time and minimizing the overall energy consumption.

The earliest deadline first (EDF) scheduling approach has been investigated to run real-time tasks and to place processes in a priority queue. High priority is assigned for packets closest to their deadline or expiry in the queue [140–142]. Houben, et al. [142] have considered minimizing energy depletion in real-time systems by sorting the tasks with enhanced EDF to vary the processor modes determined by supply voltage, frequency and performance requirements. However, several request packets in the queue can have the same deadline, and the EDF algorithm does not consider the time redundancy management of these packets, nor does it give priority to packets that come from longer distances, with more hops which causes higher energy depletion. Gomathi and Mahendran [143] implemented the nearest job next (NJN) scheduling method. The NJN protocol always chooses the nearest sensor node as the first node to gather and deliver its data to the final destination, which helps the system to reduce latency.

An approach suggested by [144] proposed an energy-efficient heterogeneous dual core processor for IoT devices. The proposed scheme included a fast CoreH and an ultra-low-power CoreL processor. This technique assigns the tasks between these two processors and performs multi-tasks at the same time. However, the problem with the multi-core processor system is that requires a large memory to hold the packets during processing. Additionally, overheating due to the use of two processors can cause the damage to the device [145].

Surveys such as that conducted by Ricardo et al. [146] have introduced a good survey paper on some specific category of duty-cycling mechanism in WSNs. The paper also summarized important directions and challenges that duty cycling will face with new emerging WSNs. Another study [147] revealed a new forwarding method for WSNs based on a simple splitting procedure able to enhance the lifetime of the network. The paper investigates a trade-off between energy efficiency and reliability of the proposed forwarding scheme when duty-cycling techniques are considered too. This paper organizes the most important proposals into a taxonomy and provides insights into their strengths and weaknesses in relation to important characteristics of applications, mote's hardware and network deployments. The authors in [148] proposes a new scheduling algorithm for IEEE 802.15.4e MAC protocol is called time synchronized channel hopping (TSCH). This approach fits for multihop WSNs based on the highway addressable remote transducer. The proposed scheme promotes wireless sensor networks by improving low latency and duty cycle and thus utmost power efficiency.

With the density of nodes increases, scheduling different types of data packets such as low or high priority data at CH nodes is essential for decreasing energy depletion, capacity and end-to-end delay. For example, if the queue is not prioritized over others, then the packets drop might happen and retransmission is required. This problem can be serious

for border nodes which access many devices and use several hops to achieve the required target. This means that these nodes consume more energy than other nodes in the network located close to the destination. Therefore, scheduling at CH nodes which gives priority to the data that comes from further nodes is crucial for energy-saving.

### 3.7.3. Aggregation Methods

The integration of WSNs and IoT elements is all about connecting devices to the Internet, making them more convenient to use and maximizing their efficiency [149]. Such integration can generate a large volume of data by these smart devices. There is considerable data redundancy in such networks due to dense deployment. Redundant data requires a considerable amount of energy to process and transmit [150]. Since each sensor is provided with only limited power eliminating data redundancy would considerably improve energy consumption overall for IoT networks.

Several studies have focused on reducing the number of data volume and data transmissions for sensor networks. A clustering algorithm is introduced in [80,151] to minimize energy depletion and lengthen the network lifetime. The proposed method partitions the sensing field into some cells and each cell elects single node to act as a cell head for all of them. Thus, one node forwards all data collected to the cell head node, which accepts the data from their associated nodes, eliminates redundant data, and then the remaining data are delivered to the final destination.

In [150], an energy-efficient in-network RFID data filtering scheme (EIFS) was proposed. The algorithm divided the sensing field into subgroups and each group has a single CH node. The CH node removes duplicated packets from its associated nodes and forwards the filtered packets to the final destination. In another study [152], the authors suggested a technique that reduced the number of data transmissions, whereby the proposed method controlled the RF-transmit operation. ON/OFF begins only when the data sensed was largely different from the previous state. In other words, the RF did not send data to the ultimate receiver if the current value is approximately same as the last recorded value.

Recent studies have confirmed that cloud computing technology offers several advantages to WSNs and IoT in terms of scalability, storage, computing tasks, etc. over the Internet. It can be used to analyze the data gathered and disseminated by sensors and IoT devices [39]. For example, Vincent, et al. [39] investigated a cloud-based architecture to enable data gathering, processing and monitoring for IoT devices. The proposed system gathers the data from various IoT devices and forwards it to the end-users via cloud infrastructure. This study aimed to provide interoperability and an efficient communication mechanism for IoT devices. The study by [153] implemented a real-time monitoring system for soil nutrient using WSN. In the proposed system, sensors measured the macro-nutrient of soil and transmitted the information to the cloud infrastructure. The user can access this information and monitor the field conditions from anywhere via a website.

Several studies have highlighted the need for real-time monitoring that can gather the data from IoT devices based on the message queuing telemetry transport (MQTT) protocol. MQTT is widely applied in IoT due to its low-overhead protocol that emphasizes the bandwidth and processor limitations of the IoT devices. It uses publish/subscribe pattern and translates messages between sensors, devices, servers and applications [24]. MQTT with IoT has been used in many applications such as military, agriculture, retail, healthcare, environmental monitoring, industry, etc. In [154], the authors proposed a network of IoT monitoring devices for fire detection. The proposed system was able to detect and monitor fires and send the information to the concerned people and authorities so that preventive measures could be taken. In [155], the authors used sensors to measure body temperature, pulse rate, body movement of patients and this measured data were uploaded to the MQTT server. The proposed system aimed to help the doctors to monitor patients from any location and at any time. It also helped patients to view and check their health condition remotely. Another study by [156] proposed a web-based interface for controlling

and monitoring an arm robot. The proposed system gives low latency data transmission via using MQTT protocol.

## 4. Aspects of Energy-Efficiency Optimization Methods

Energy-saving is one of the most essential requirements for WSNs and IoT networks, as batteries are usually the main source of power for these networks. Suitable techniques can reduce energy consumption and extend network lifetime. Thus, different techniques in different aspects of IoT-based WSNs are required to minimize energy consumption. According to an investigation by [157], authors introduced elliptic curve cryptography-based mutual authentication (EMA) and capability-based access control (CBAC) model to enhance mutual authentication and reduce energy consumption against other. The authors [158] also presented energy consumption analysis of lightweight cryptographic algorithms. The proposed method is more effectively for secure IoT devices with low power consumption. In addition, The authors showed that design of efficient power converters can also reduce the energy consumption of IoT devices. For example, authors [159] investigated the designed DC–DC converter adapted for ultra-low-power operating for IoT applications. The proposed method is low-power, small area and high resolution DPWM design that prepares for DC–DC converter to power the ULV operating IoT networks. Energy-efficient machine learning on the edges for IoT devices were carried out by [160]. The proposed algorithm has shown up to 14.46% improvement in energy consumption. In a different study, Adil et al. [161]. reported that the malicious node is also attempted to attack and deny service to other nodes in the sensing field. Thus, it drops some transmission packets from a node to others. As a result, energy is dissipated due to retransmission packets [162].

## 5. Research Gaps

The literature review demonstrated the existence of many avenues for minimizing energy wasted and lengthening the network lifetime. It presented many possibilities and identified numerous limitations. This paper has scrutinized the literature to obtain an insight into the perspectives addressed by previous researchers and the gaps left by existing solutions.

In the sensing field, sensor nodes depend on other intermediate nodes to forward their data. Some sensor nodes send their data over single or multiple hops to arrive the BS through CH nodes. The CH node can become overloaded due to the number of surrounding nodes that are connected and forwarded their data to it. The probability of retransmitting some of these packets at the CH nodes will increase. Therefore, data delivered by the sensor nodes must be prioritized at CH nodes based on the energy consumed by each packet. However, no report has been presented assigning high priority to packets that come from the furthest distances to the CH nodes. Since these packets more quickly exhaust the network resources because they require more links and nodes to reach their ultimate destination, one of our studies suggests the introduction of a novel scheduling algorithm called the long hop (LH) to optimize energy used in sensor networks [127]. The LH algorithm assigns high priority to the scheduling of packets arriving after a greater number of hops, from longer distances. These packets serve first at the CH nodes to prevent them from being retransmitted, and so conserving energy.

The literature review also introduced the energy consumed due to delivering data from the source node to the required destination. Based on the multihop concept, the transmitted packets access multiple nodes and experience a greater number of hops to reach the intended destination. Each node involved in the transmission process has many forwarding nodes. However, a higher number of forwarding nodes causes higher energy consumption. Thus, another research work introduces a new routing strategy that sends data to the next-hop node within a shorter transmission distance and fewer forwarding nodes to optimize the energy consumption. The proposed protocol evades delivering packets to nodes that have many forwarding nodes, therefore balancing the load traffic and enhancing the network performance and lifetime of the network.

There have been relatively a large and growing studies of literature reports attempts to minimize the energy depletion of sensor networks, this author believes that there are only a few research studies on theoretical energy analysis of sensor nodes based on less transmission distance, interference and the creation of CH nodes. Interference is also one of the main factors that cause data collisions and consequent energy wastage. A node with fewer neighbors has less overhearing and interference. Therefore, one of previous works presents a new routing method that elects the next-hop node to be one with least transmission distance and fewer neighbor nodes and thus less interference [163]. The proposed scheme also introduces a new method that selects CH nodes around a single BS based on lower transmission distances. Both techniques minimize energy depletion and lengthen the lifetime of network.

As shown in the literature, previous research focused on efficient transmission data in the WSNs and IoT applications. However, most of these studies considered either eliminating redundant data or monitoring active devices remotely. Thus, the author believes that investigating both the filtering of redundant data and the remote monitoring of the behavior and condition of the smart objects. These works will decrease the volume of data forwarded which, sequentially, will minimize the process of scheduling and routing data on each device on the network, consequently minimizing the energy consumption and success the IoT technology.

There are many published studies (e.g., [164–166]) that introduce a survey of relatively recent techniques and methods proposed for performing data aggregation in WSNs. The survey shows that improve energy consumption by reducing the transmission of redundant data and thus lengthen the network lifetime. In [167], authors have demonstrated a new protocol called Energy-efficient and balanced cluster-based data aggregation algorithm (EEBCDA). The method aims to overcome the energy dissipation issue in the cluster-based aggregation data. In this protocol, the network is classified into rectangular grids of unequal size and the CH node keeps on rotating in each cluster. The CH node in each grid is picked based on the high residual energy. The EEBCDA method balances energy usage in each grid and thus prolong the network lifetime. Researchers reviewed the available literature on the clustering–based aggregation protocol approach in WSN for minimum communications and extend the network lifetime [165]. Researchers have studied and compared some of the state-of-the-art data-gathering techniques considering their trade-off between reliability (i.e., packet loss and reconstruction error) and energy consumptions (i.e., network lifetime) by taking into account both compression and networking aspects [166]. The paper summarizes the results as follows: (i) Distributed source coding (DSC) and transforms and encoding compression (TEC) techniques should be preferred for prolonging the lifetime of the network. (ii) Compressive sensing (CS) should be preferred when high reliability is needed. (iii) Chinese remainder theorem (CRT) should be preferred for its inherent low complexity. However, to the best of the author's knowledge, there is no best solution for all possible applications and that only the trade-off between reliability, energy consumptions and complexity can drive the choice of the data-gathering technique to be used for a specific application.

Finally, these methods can broadly be classified into several characteristics and summarized in Table 3, thus, the table presents a summary of previous methods surveyed in this paper.

**Table 3.** A summary of previous methods surveyed in this paper.

| Protocols | Mobility | Hop Limit | Use of Location Info. | Type of Protocol | Network Improvement | Selected CH Node | Ref. |
|---|---|---|---|---|---|---|---|
| LEACH | Fixed | Single hop | No | Routing | Energy-efficiency | Randomly | [77] |
| LEACH-C | Fixed | Single hop | Yes | Routing | Energy-efficiency | A node with more energy in a cluster | [84] |
| E-LEACH | Fixed | Multi-hops | Yes | Routing | Energy-efficiency | A node with the highest remaining energy | [85] |
| EAMMH | Fixed | Multi-hops | Yes | Routing | Energy-efficiency | The main CH should be an optimum distance from these child-CH nodes. | [86] |
| EAMR | Fixed | Multi-hops | Yes | Routing | Energy-efficiency | A node is selected a CH node until its energy falls below a threshold value. | [87] |
| HEED | Fixed | Single hop & Multi-hops | Yes | Routing | Energy-efficiency | The selected CH node based on the high residual energy | [82] |
| H-HEED | Fixed | Single hop & Multi-hops | Yes | Routing | Energy-efficiency | The H-HEED finds the center of each cluster and then allocates the node nearest the cluster center as a CH. | [90,91] |
| ER-HEED | Fixed | Single hop & Multi-hops | Yes | Routing | Energy-efficiency | A node with the highest level of energy is a CH. | [92] |
| E-HEED | Fixed | Multi-hops | Yes | Routing | Energy-efficiency | The CH nodes according to the least transmission distance from the BS. | [93] |
| PEGASIS | Fixed | Multi-hops | No | Routing | Energy-efficiency | Randomly | [83] |
| DORA | Fixed | Multi-hops | No | Routing | Energy-efficiency | Randomly | [98] |

**Table 3.** *Cont.*

| Protocols | Mobility | Hop Limit | Use of Location Info. | Type of Protocol | Network Improvement | Selected CH Node | Ref. |
|---|---|---|---|---|---|---|---|
| LLND | Fixed | Multi-hops | No | Routing | Interference and channel fading | Fixed | [100] |
| AS-MAC | Fixed | Single hops | No | Scheduling | Energy-efficiency | Fixed | [101] |
| Centralized range-based localization | Fixed | Multi-hops | No | Routing | Energy-efficiency and network delay | — | [105] |
| DEE | Fixed | Multi-hops | No | Routing | Energy-efficiency and reducing unnecessary traffic | The CH nodes is selected according to the ratio between remaining energy of a node and distance | [106] |
| OLMS | Moved | Multi-hops | Yes | Routing | Energy-efficiency | The best location of the MS based on the minimum energy cost for data delivery of CHs | [108] |
| TBMS | Moved | Multi-hops | Yes | Routing | Energy-efficiency | The CH is closed node to the BSs | [61] |
| EIGR | Fixed | Multi-hops | No | Routing | Reducing energy consumption and interference | — | [112] |
| TH-SS | — | Multi-hops | No | Routing | Energy harvesting from interference | — | [114] |
| LPWAN | — | Multi-hops | No | Routing | long range communication and low power consumption | — | [118] |
| Handling Failures of Static Sensor Nodes | Fixed and Moved | Multi-hops | No | Routing | Energy-efficiency | — | [116] |

**Table 3.** *Cont.*

| Protocols | Mobility | Hop Limit | Use of Location Info. | Type of Protocol | Network Improvement | Selected CH Node | Ref. |
|---|---|---|---|---|---|---|---|
| BCH | Fixed | Multi-hops | No | Routing | Energy-efficiency | Create backup CH node for the original one | [119] |
| DSP | — | Multi-hops | No | Routing | The network traffic performance | — | [122,123] |
| CHRA | Fixed | Multi-hops | No | Routing | Energy-efficiency and network traffic performance | Randomly | [124] |
| EERH | Fixed | Multi-hops | No | Routing | Energy-efficiency | — | [127] |
| GCEEC | Fixed | Multi-hops | No | Routing | Reduced data traffic and Energy-efficiency | The CH node is selected from the centroid position | [129] |
| AODV | — | Multi-hops | No | Routing | Energy-efficiency | — | [125] |
| DSR | — | Multi-hops | No | Routing | Energy-efficiency | — | [126] |
| SPT | Fixed | — | No | Scheduling | Enhancing service response time and minimizing the overall energy consumption. | Randomly | [120] |
| EDF | Fixed | — | No | Scheduling | Energy-efficiency | — | [134–136] |
| NJN | — | Single hop | No | Scheduling | Reduce latency | — | [137] |
| Dual core processor | — | Single hop | No | Scheduling | Reduce latency and Energy-efficiency | — | [138] |

**Table 3.** *Cont.*

| Protocols | Mobility | Hop Limit | Use of Location Info. | Type of Protocol | Network Improvement | Selected CH Node | Ref. |
|---|---|---|---|---|---|---|---|
| TSCH | Fixed | Multi-hops | No | Routing | Improving low latency and duty cycle and thus utmost power efficiency. | — | [142] |
| Clustering algorithm | Fixed | Multi-hops | No | Routing | Energy-efficiency | Randomly | [145] |
| EIFS | Fixed | Multi-hops | No | Routing | Energy-efficiency | Randomly | [144] |
| LH | Fixed | Multi-hops | No | Scheduling | Energy-efficiency | The CH is closed node to the BS. | [121] |
| SPLL | Fixed | Multi-hops | No | Routing | Energy-efficiency | The CH is closed node to the BS. | [46] |
| EEBCDA | Fixed | Multi-hops | No | Routing | Energy-efficiency | The CH node keeps on rotating in each cluster. | [155] |

## 6. Conclusions

IoT is having a major impact on the digital world and how we interact with the Internet. WSNs are a key enabling technology for the IoT. Sensor nodes capable of detecting the required information, performing some processing and communicating with other connected nodes are the main component of these networks. However, the life of these nodes is often restricted by being powered by a battery with a limited life, constraining processing ability, memory, and radio communications. Energy efficiency is one of the most crucial issues for WSN; it is not rational to consume energy on protocol overheads, the transmission of unneeded data or non-optimized transmission of data packets, especially retransmissions, due to inefficient scheduling and routing algorithms. Hence, the main aim of any energy-efficient strategy is to keep the sensor nodes alive for longer and thus lengthen the network lifetime. Therefore, in this survey paper, our goal is to introduce the research trends and recent work on the use of IoT technology, key enabling technologies, various sources of energy wastage and different solutions have been mentioned in the literature. By discussing the motivational factors, we have clarified several challenges that need to be addressed to enable theoretical and practical implementations of WSN-based IoT networks.

**Author Contributions:** Conceptualization and methodology, L.F.; part supervision, W.G. and L.A.; investigation and original draft preparation, R.S.H., A.S.A., A.H.F., experiment and writing assistance, M.A.F. and M.A.-A. All authors have read and agreed to the published version of the manuscript.

**Funding:** This research received no external funding.

**Informed Consent Statement:** Informed consent was obtained from all subjects involved in the study.

**Conflicts of Interest:** The authors declare no conflict of interest.

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
