# Peer review of "Energy Efficiency for Green Internet of Things (IoT) Networks: A Survey†"

_2673-8732, doi:10.3390/network1030017_

Round 1
Reviewer 1 Report
This survey is almost overlapped with the published thesis (the link is attached). I don't think it's necessary to publish another paper with very little extended information. The contribution is very limited.
https://e-space.mmu.ac.uk/626319/
Author Response
Reviewer 1
This survey is almost overlapped with the published thesis (the link is attached). I don't think it's necessary to publish another paper with very little extended information. The contribution is very limited.
We thank you for your comment on the overall paper. We completely agree that the paper is part of the thesis, however, we have updated the paper more than 40%. We have included different sections, challenges, information, figures, tables, etc. This could help and support future researchers in their articles and knowledge.
Reviewer 2 Report
The authors have presented a survey of various energy-efficiency techniques which have been proposed in past and recent literature in the context of wireless sensor networks (WSNs), especially applying to the Internet of Things (IoT).
Sections 1-3 provide a very good introduction to the motivation behind this study as well overview of both IoT and WSNs. Section 4 is the main survey and Section 5 discusses some possible future directions. While Section 4 is well-written, there are still some concerns that must be addressed. Please find detailed comments below:
Although the manuscript mentions "IoT" in the title and also several times in the main text, the current survey is mostly focused on the network / communication component of IoT. The discussion in Section 4 is regarding different routing protocols, communication overheads and scheduling mechanisms. The current discussion in Section 4 includes mostly the communication component, which has already been surveyed in similar ways in previous work. The IoT concept also includes other important components such as data sensing, data processing, security and power supply requirements. While these terms are briefly mentioned in Section 3, they are not surveyed at all. This survey will greatly improve in quality if these aspects are also included, e.g., different circuit techniques for low-power sensing and analog-to-digital conversion, computing at the IoT node using low-power micro-processors (e.g., ARM, RISC-V), energy-efficient custom hardware accelerators for machine learning, signal processing, encryption (e.g., AES), authentication (e.g., elliptic curve cryptography, transport layer security), design of efficient power converters, energy harvesting techniques, wake-up radios, etc.
The organization of Section 4 should be improved - the sub-sections should be written following the classification and ordering in figure 13.
Please rotate figure 13 by 180 degree for better readability.
page 25-27 - please rotate the tables by 180 degree for better readability
page 3 - "advertised by APPLE, SAMSUNG, etc." - please include corresponding references
Author Response
Thanks for your positive comment on overall paper. We have effectively addressed the pointed issues (below) towards enhancing the quality of the paper.
1) The IoT concept also includes other important components such as data sensing, data processing, security and power supply requirements. While these terms are briefly mentioned in Section 3, they are not surveyed at all. This survey will greatly improve in quality if these aspects are also included, e.g., different circuit techniques for low-power sensing and analog-to-digital conversion, computing at the IoT node using low-power micro-processors (e.g., ARM, RISC-V), energy-efficient custom hardware accelerators for machine learning, signal processing, encryption (e.g., AES), authentication (e.g., elliptic curve cryptography, transport layer security), design of efficient power converters, energy harvesting techniques, wake-up radios, etc.
We thank you for your comment on reference list. We completely agree and understand the significance of recent references in the quality of publications. To incorporate the suggestion, we have included different aspects of IoT that could reduce the energy consumption. See (page 18, lines 520 -532), (page 20, lines 605 – 610), (page 20, lines 635 – 641), (for scheduling algorithms, turn ON/OFF the antenna, see section 3.7.2), (for sensing and gathering data see 3.7.3. Aggregation Methods). We have also written new section, that included different aspects of energy efficiency optimization methods, see section 4. Page 24.
2) The organization of Section 4 should be improved - the sub-sections should be written following the classification and ordering in figure 13.
We thank you for your comment on organization of Section 4. We completely agree and understand to orginaze section 4 into sub-sections, To incorporate the suggestions, (see subsections 3.7.1, 3.7.2, 3.7.3, pages 14, 21, 23).
3) Please rotate figure 13 by 180 degree for better readability.
Thanks for your comment on figure direction. We are extremely sorry for the representational mistake. To incorporate the suggestions, ( see page 13)
4) page 25-27 - please rotate the tables by 180 degree for better readability
Thanks for your comment on tables directions. We are extremely sorry for the representational mistake. To incorporate the suggestions, (see pages 26, 27, 28)
5) page 3 - "advertised by APPLE, SAMSUNG, etc." - please include corresponding references
Thank you for your precise comment on reference list. We have updated the reference (see line 100, Ref. 25).

Reviewer 3 Report
This paper aimed to develop energy-efficiency technique for WSNs that enable the IoT. Also, it reviewed the literature that discussed the most relevant methods to minimising the energy exhaustion of IoT and WSNs such as path selection, method scheduling data method and efficient data aggregation. This paper is an interesting and presents the relationship between WSNs and IOT in terms of energy consumption. However, this can be considered for publication after performing the following comments:
1) Abstract Section need to revise. The authors need to include the research findings and the research importance and contribution in order to easy for readers and other researchers.
2) In the Introduction section, no need for Figure 1. My suggestion is to delete it and the paragraph that explain it.
3) In addition, all statistical data about the usage of IOT in the Introduction section should be moved to new section of (Literature review).
4) Figures 2,3, 4 and 5 move to section Literature review.
5) In Lines 108-113 the author claimed "This means it is not rational to waste energy on protocol over heads, the transmission of unneeded data or non-optimised transmission of data packets, especially retransmissions, due to inefficient scheduling and routing algorithms. Thus, it would be prudent to design and implement systems that are designed to minimise energy consumption, and so increase the node lifetime and thus the life of the overall network. This claim need justification from authors. I suggest them to include the following
- An efficient load balancing scheme of energy gauge nodes to maximize the lifespan of constraint oriented networks.
- MAC-AODV based mutual authentication scheme for constraint oriented networks.
6) I suggest for authors to add new section "Motivation of the Research".
7) In section 3, Overview of WSNs, this section is limited from to very little studies that focused on energy proficient load balancing routing scheme for wireless sensor networks, Thus, I suggest for authors to includes the following
- An energy proficient load balancing routing scheme for wireless sensor networks to maximize their lifespan in an operational environment.
- Improving energy efficiency with content-based adaptive and dynamic scheduling in wireless sensor networks.
8) In conclusion section, the authors need to clarify the contribution of the research and its importance in terms of theoretical and practical.
Author Response
Reviewer 3
This paper aimed to develop energy-efficiency technique for WSNs that enable the IoT. Also, it reviewed the literature that discussed the most relevant methods to minimising the energy exhaustion of IoT and WSNs such as path selection, method scheduling data method and efficient data aggregation. This paper is an interesting and presents the relationship between WSNs and IOT in terms of energy consumption. However, this can be considered for publication after performing the following comments:
1) Abstract Section need to revise. The authors need to include the research findings and the research importance and contribution in order to easy for readers and other researchers.
Thank you very much for your precise comment on the abstract. To incorporate the suggestion, we have updated the abstract (see abstract, lines, 13-16)
2) In the Introduction section, no need for Figure 1. My suggestion is to delete it and the paragraph that explain it.
Thank you for your precise comment on Figure 1. We have removed Figure 1 from the paper.
3) In addition, all statistical data about the usage of IOT in the Introduction section should be moved to new section of (Literature review).
4) Figures 2,3, 4 and 5 move to section Literature review.
Thank you very much for your comment on the literature review. To incorporate the suggestion, we have moved the statistics data and figures to the literature review section. (see Section 3, Literature Review).
5) In Lines 108-113 the author claimed "This means it is not rational to waste energy on protocol over heads, the transmission of unneeded data or non-optimised transmission of data packets, especially retransmissions, due to inefficient scheduling and routing algorithms. Thus, it would be prudent to design and implement systems that are designed to minimise energy consumption, and so increase the node lifetime and thus the life of the overall network. This claim need justification from authors. I suggest them to include the following
- An efficient load balancing scheme of energy gauge nodes to maximize the lifespan of constraint-oriented networks.
- MAC-AODV based mutual authentication scheme for constraint oriented networks.
We thank you for your comment on reference list. We completely agree and understand the significance of recent references in the quality of publications. To incorporate the suggestion, the pointed references are cited (see lines 55-61, Ref. (11, 13)).
6) I suggest for authors to add new section "Motivation of the Research".
Thank you for your precise comment on Motivation of the Research. We have written a new section of Motivation of the Research, (see page 2, section 2. Motivation of the Research).
7) In section 3, Overview of WSNs, this section is limited from to very little studies that focused on energy proficient load balancing routing scheme for wireless sensor networks, Thus, I suggest for authors to includes the following
- An energy proficient load balancing routing scheme for wireless sensor networks to maximize their lifespan in an operational environment.
- Improving energy efficiency with content-based adaptive and dynamic scheduling in wireless sensor networks.
Thank you for precisely pointing out new references. We have incorporated the suggestion by adding the suggested references (see lines 242 – 252).
8) In conclusion section, the authors need to clarify the contribution of the research and its importance in terms of theoretical and practical.
Thank you for your comment on conclusion section. To incorporate the suggestion, we have updated the conclusion (see lines 919 - 924).

Round 2
Reviewer 2 Report
The manuscript has been revised according to previous comments.
Author Response
Thank you very much for your precise comment on the overall paper.
Reviewer 3 Report
The author still need to add more justification from recent studies in WSN as following
- A new scheme for detecting malicious attacks in wireless sensor networks based on blockchain technology
- An anonymous channel categorization scheme to detect jamming attacks in wireless sensor networks
- After minor correction can be accepted the paper
Author Response
We thank you for your comment on reference list. We completely agree and understand the significance of recent references in the quality of publications. To incorporate the suggestion, see Page 24, lines 833-836